# A holistic approach for suppression of COVID-19 spread in workplaces and universities

**Sarah F. Poole**[1]*, **Jessica Gronsbell**[1], **Dale Winter**[1], **Stefanie Nickels**[1], **Roie Levy**[1], **Bin Fu**[1], **Maximilien Burq**[1], **Sohrab Saeb**[1], **Matthew D. Edwards**[1], **Michael K. Behr**[1], **Vignesh Kumaresan**[1], **Alexander R. Macalalad**[1], **Sneh Shah**[1], **Michelle Prevost**[1], **Nigel Snoad**[1], **Michael P. Brenner**[2], **Lance J. Myers**[1], **Paul Varghese**[1], **Robert M. Califf**[1], **Vindell Washington**[1], **Vivian S. Lee**[1], **Menachem Fromer**[1]

1 Verily Life Sciences, South San Francisco, CA, United States of America, 2 Google Research, Mountain View, CA, United States of America

\* sarahfpoole@verily.com

## Abstract

As society has moved past the initial phase of the COVID-19 crisis that relied on broad-spectrum shutdowns as a stopgap method, industries and institutions have faced the daunting question of how to return to a stabilized state of activities and more fully reopen the economy. A core problem is how to return people to their workplaces and educational institutions in a manner that is safe, ethical, grounded in science, and takes into account the unique factors and needs of each organization and community. In this paper, we introduce an epidemiological model (the "Community-Workplace" model) that accounts for SARS-CoV-2 transmission within the workplace, within the surrounding community, and between them. We use this multi-group deterministic compartmental model to consider various testing strategies that, together with symptom screening, exposure tracking, and nonpharmaceutical interventions (NPI) such as mask wearing and physical distancing, aim to reduce disease spread in the workplace. Our framework is designed to be adaptable to a variety of specific workplace environments to support planning efforts as reopenings continue. Using this model, we consider a number of case studies, including an office workplace, a factory floor, and a university campus. Analysis of these cases illustrates that continuous testing can help a workplace avoid an outbreak by reducing undetected infectiousness even in high-contact environments. We find that a university setting, where individuals spend more time on campus and have a higher contact load, requires more testing to remain safe, compared to a factory or office setting. Under the modeling assumptions, we find that maintaining a prevalence below 3% can be achieved in an office setting by testing its workforce every two weeks, whereas achieving this same goal for a university could require as much as fourfold more testing (i.e., testing the entire campus population twice a week). Our model also simulates the dynamics of reduced spread that result from the introduction of mitigation measures when test results reveal the early stages of a workplace outbreak. We use this to show that a vigilant university that has the ability to quickly react to outbreaks can be justified in implementing testing at the same rate as a lower-risk office workplace. Finally, we quantify the devastating impact that an outbreak in a small-town college could have on the

**Data Availability Statement:** The majority of the relevant data are within the manuscript and its Supporting Information files. The manuscript refers to some publicly available data, which is available

at https://covid19-projections.com/ and https://rt.live/.

**Funding:** The authors received no specific external funding for this work.

**Competing interests:** At the time of work, SFP, JG, DW, SN, RL, BF, MB, SS, MDE, MKB, VK, ARM, SS, MP, NS, LJM, PV, RMC, VM, VSL, and MF were employees of and owned equity in Verily Life Sciences. While this manuscript does not explicitly mention Verily's Healthy At Work program, the models presented herein are used in that program.

surrounding community, which supports the notion that communities can be better protected by supporting their local places of business in preventing onsite spread of disease.

## Introduction

The COVID-19 pandemic is a global crisis, with a devastating impact on people, organizations, and industries across the world. Efforts to reignite economic activity require a robust and safe return-to-work strategy. The signs and symptoms that characterize the disease vary, the mechanics of immunity are not fully understood, and a vaccine is still not available in most parts of the world [1]. Additionally, a large proportion of infected individuals may never experience symptoms and can silently spread the disease [2]. Therefore, an approach based solely on symptom tracking and testing of symptomatic individuals will be insufficient to prevent spread in most circumstances. Instead, augmenting symptom-based testing with cost-effective monitoring testing of the workforce has been proposed as a more promising strategy [2].

Ideally, an employer must consider several factors when selecting a testing strategy. The disease prevalence in the surrounding community, and the rate of change of this prevalence, will impact the prevalence among employees and should thus be accounted for. Furthermore, the choice of testing strategy should incorporate features of the workplace such as the degree of close-contact interactions between employees and the amount of time that employees spend at work. In addition to selecting testing strategies for symptomatic and asymptomatic individuals, employers must make choices about how many employees they will bring back to work, and they must also consider the requirements for employees who test positive, such as the amount of time that they are asked to self-isolate away from the workplace.

These numerous considerations highlight the need for models that enable employers to anticipate, explore, and decide on policies that are appropriate for the particulars of their workplace. Such models can present the projected impact of various testing strategies, allowing an employer to make an informed decision on the most appropriate strategy. Models that give these insights have been explored in a university setting [3–7] and in a healthcare setting [8] but have not been thoroughly explored across different workplace settings.

An important component of virus spread in a workplace is the level of interaction of employees with non-employees in the community. This consideration is also important in a university setting, although since college campuses are often relatively self-contained a model may choose to ignore the ongoing influence of the community. Lopman et al. [6] and Lyng et al. [7] capture the impact of the community by including a continuous rate of spontaneous infection in the university population. Paltiel et al. [4] instead add regular exogenous 'shocks' of infection to the university population to simulate the impact of the community, while Gressman et al. [5] include a 25% chance that one member of the university population becomes spontaneously infected each day. However, none of these approaches are able to capture the time-varying impact of a community that is undergoing an outbreak and are also not able to capture the impact of the workplace on the community.

We present here a novel compartmental epidemiological model that accounts for SARS-CoV-2 transmission both within the workplace and in the surrounding community. This model is intended for use in forecasting prevalence in a workplace and guiding its choice of testing strategy. This model is designed to simulate how testing can be used alongside education and other workplace nonpharmaceutical interventions (NPI), such as masking policies, increased spacing of desks, and staggered return-to-work schedules, to allow workplaces to

resume on-site activities while minimizing the risk of a new outbreak. Note that, in this paper, we use the term "outbreak" to refer to out-of-control spread of the virus, rather than a specific number of infected cases. We apply this model to investigate disease dynamics upon reopening of various workplace and university environments, demonstrating the flexibility of our approach in understanding disease spread and devising testing plans.

## Materials and methods

We leverage a dynamic, deterministic, two-group thirteen-compartment model (Fig 1), which contains a SEPAYR (Susceptible—Exposed—Presymptomatic—Asymptomatic—sYmptomatic—Recovered) model for non-employees ("community"), alongside a SEPAYDR (Susceptible—Exposed—Presymptomatic—Asymptomatic—sYmptomatic—Detected—Recovered) model for employees ("workplace"). This "Community-Workplace" model accounts for transmission dynamics within and between the workplace and the community, and it can be used to simulate disease dynamics and inform the selection of a testing strategy for a specific workplace. For full details on the model and its parameters, please refer to S1 File. For details on how model parameters are chosen, see S2 File.

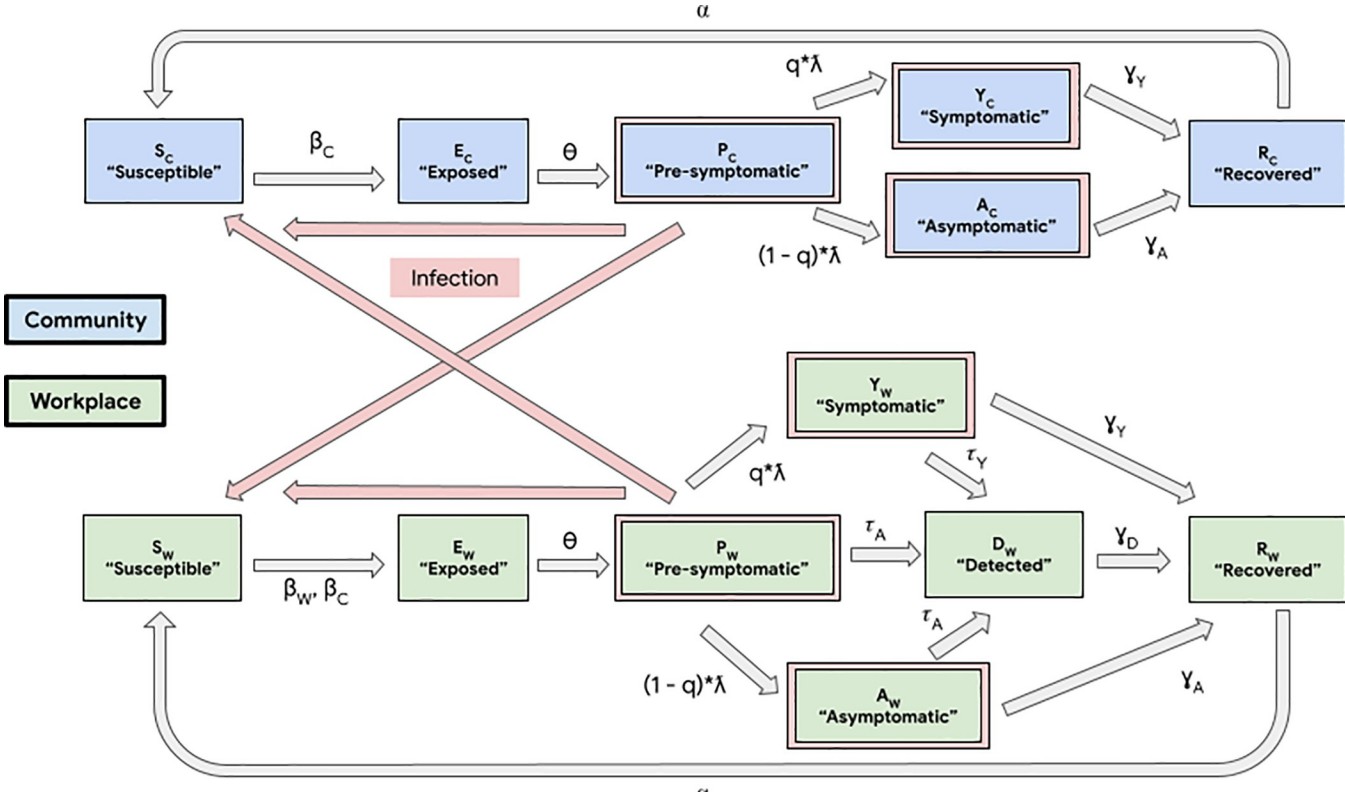

**Fig 1. "Community-Workplace" compartmental model of disease spread in the workplace and community.** The non-employees ("Community", shaded blue, denoted with subscript "C") are modeled using SEPAYR (Susceptible—Exposed—Presymptomatic—Asymptomatic—sYmptomatic—Recovered) compartments. The employees ("Workplace", shaded green, denoted with subscript "W") additionally can move through a Detected compartment, resulting in a SEPAYDR (Susceptible—Exposed—Presymptomatic—Asymptomatic—sYmptomatic—Detected—Recovered) model that tracks the stages of COVID-19 infection and detectability by workplace testing. Compartments of individuals that are sources of infections are outlined in pink, and pink arrows denote the paths of potential infections, i.e., disease transmissions. Model transition rate parameters are denoted on compartment-to-compartment transition arrows, and their semantics are detailed in S1 Table.

To demonstrate the broad applicability of our modeling approach in the real world, we examined three case studies capturing some of the diversity of businesses and institutions of higher education in the United States:

a. Office workplace (representing a "9-to-5" workplace with lower density / contact load)

b. Factory floor (representing a "9-to-5" workplace with higher density / contact load)

c. University campus (representing an institution where many of the population spend a majority of their time, including sleeping, and where the population experiences higher density / contact load)

There are two key model parameters that are varied to emulate the environment for each case study. These are the basic virus reproduction number (i.e., the mean number of people in a fully susceptible population that are infected with SARS-CoV-2 by a single infected person) in the workplace ($R0_W$), and the proportion of time employees spend at work and interacting only among themselves (p). Note that all parameters not used to capture this variation across the different environments were held constant across all case studies herein (see S2 Table), except where noted below.

An $R0_W$ of 3 was used to simulate an indoor "Office workplace" with a medium burden of employee-employee interactions, along with a value of p of 33%. A higher $R0_W$ of 4 was used to simulate a "Factory floor" to capture the higher interaction between employees, due in part to increased physical density. As in the "Office workplace", a value of p of 33% was used. An $R0_W$ of 4 was used to simulate a "University", to capture the heightened level of interaction expected between students who are living and attending classes together, as well as socializing after school hours. To capture the higher amount of time that on-campus students spend in one another's company, a value of p of 70% was used.

For each case study scenario, a range of testing strategies were simulated and compared. Modeling a variety of testing strategies assists in the decision-making process, by yielding insights into the potential impact of the different strategies [9]. The strategies investigated in our case studies, ordered by increasing testing volume, are as follows:

- **NO TESTING**: No testing at all.

- **INITIAL TESTING ONLY (I)**: Initial testing only ("back to work testing").

- **I + SYMPTOMATIC TESTING (S)**: Initial testing, along with testing any employee that develops and self-reports symptoms, i.e., all symptomatics (see S2 File for details on the choice of parameter defining the proportion of cases that develop symptoms).

- **I + S + TEST 5% OF ASYMPTOMATICS EVERY WORK DAY**: Initial testing, testing of all symptomatics, and testing of a randomly selected 5% of asymptomatic individuals each "work day" (i.e., 5 days a week). This strategy results in all asymptomatics being tested approximately once every four weeks.

- **I + S + TEST 10% OF ASYMPTOMATICS EVERY WORK DAY**: Initial testing, testing of all symptomatics, and testing of a randomly selected 10% group of asymptomatic individuals each work day (i.e., 5 days a week). This strategy results in all asymptomatics being tested approximately once every two weeks.

- **I + S + TEST 20% OF ASYMPTOMATICS EVERY WORK DAY**: Initial testing, testing of all symptomatics, and testing of a randomly selected 20% of asymptomatic individuals. This strategy results in all asymptomatics being tested approximately once every week.

- **I + S + TEST 40% OF ASYMPTOMATICS EVERY WORK DAY**: Initial testing, testing of all symptomatics, and testing of a randomly selected group of 40% of asymptomatic individuals. This strategy results in all asymptomatics being tested approximately twice per week.

A key feature of epidemiological models such as the one described here is that each of the variables of interest, in particular the number of individuals in each compartment, is tracked throughout the simulation. This permits the user to calculate and monitor a variety of metrics that can assess the projected severity and impact of COVID-19 outbreaks. For simplicity, we focus here on the prevalence of cases of active infection among employees (1.15 Equation of S1 File). This metric captures the simultaneous impact of infected employees who are detected by testing and must miss work, alongside infectious employees at work that form a pool of active risk of exposing other employees to SARS-CoV-2.

## Results

Fig 2 shows the time-based trajectories of infection prevalence in the workplace population for the corresponding case studies, generated using the "Community-Workplace" model. Other related metrics of interest are depicted in S3 File, including peak workplace prevalence (S4 Fig in S3 File), cumulative workplace prevalence (S5 Fig in S3 File), cumulative community prevalence (S6 Fig in S3 File), and total number of workplace tests conducted (S7 Fig in S3 File).

The "Office workplace" and "Factory floor" case studies (Fig 2A and 2B) differ only by the value of the parameter related to the transmission rate between employees in the workplace, $R0_W$. Comparing these case studies, we see that increasing the transmission rate within the workplace ($R0_W = 3$ in the "Office workplace" vs. $R0_W = 4$ in the "Factory floor") leads to higher peak prevalences (S4a and S4b Fig in S3 File), as well as higher cumulative prevalences, i.e., total employees infected (S5a and S5b Fig in S3 File). However, we find that increasing the testing volume diminishes these differences between workplaces. Specifically, when only symptomatic testing is conducted, the peak workplace prevalence in the two scenarios varies by 0.9%. When monitoring testing of 10% of asymptomatic people per day is added, the peak

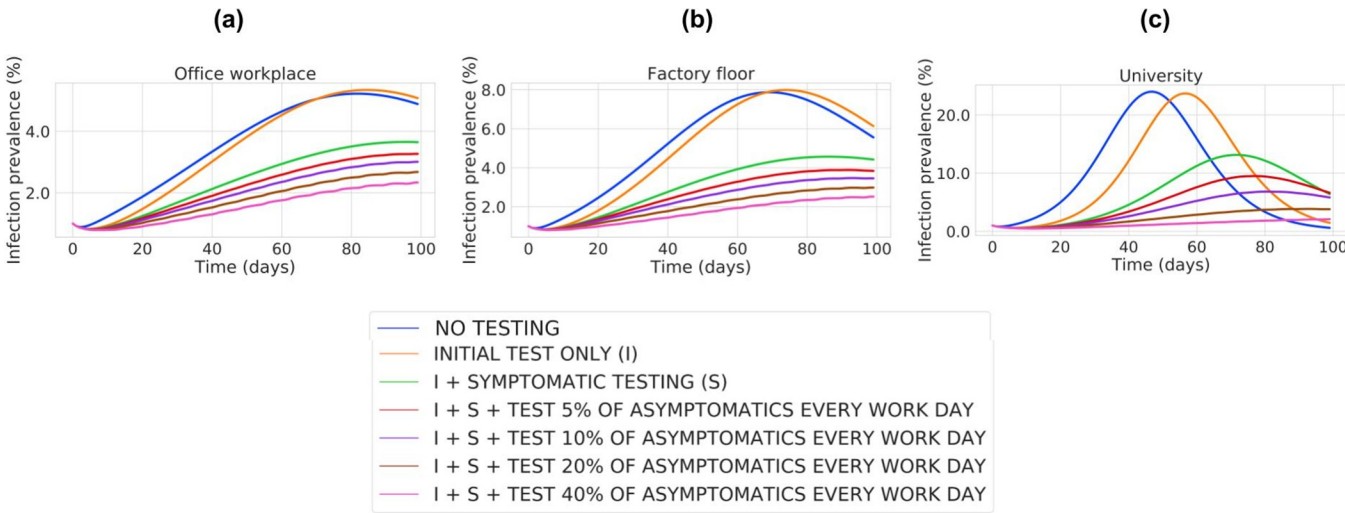

**Fig 2. Estimated prevalence trajectory for the three case studies, each under a range of testing strategies.** For each case study, the prevalence of employee infection is plotted on the vertical axis (as percentage of the total workplace population), tracked over time (in days) over the course of the simulation on the horizontal axis. For each scenario, the prevalence for the various assessed testing strategies are plotted as distinct curves and colored as per the legend. The parameters used for each case study are described in the main text, with more detail given in S1 File.

workplace prevalence in the two scenarios varies by 0.5%, and this difference drops to only 0.2% when 40% of asymptomatic people are tested each day.

The model includes a parameter, p, that describes the proportion of time that employees spend at work (see S1 Table). By defining the amount of time that employees spend interacting only with one another, this parameter modulates the amount of infection spread between the workplace and community populations. To understand the importance of this parameter, we compare the "Factory floor" and "University" case studies (Fig 2B and 2C), since all other parameters are held constant. Spending a higher proportion of time isolated in a high-contact workplace environment (in the "University") increases peak and total infections in the workplace/campus population (S4b and S4c Fig in S3 File), and the "University" requires significantly more testing to achieve parity with the "Factory Floor". Specifically, when testing 5% of asymptomatic individuals each work day, a peak prevalence of 3.9% is achieved in the "Factory Floor" setting, but 20% of asymptomatic individuals must be tested each work day in order to match this in the "University" setting. Of note, due to diminished interaction between the workplace and the community, the total percentage of individuals infected in the community is actually lower in the "University" case compared to the "Factory floor" case (S6b and S6c Fig in S3 File).

A simultaneous comparison of these three case studies is instructive with respect to the role of testing. It demonstrates that as more time is spent together in a high-contact workplace environment, more aggressive testing of asymptomatic individuals is required to keep infection at safe levels. As an example, consider a 3% peak prevalence in the workforce as a high but still tolerable threshold. To maintain prevalence below that level, the "Office workplace" must test 10% of asymptomatic individuals per work day, the "Factory floor" must test 20% of asymptomatic individuals per work day, and the "University" requires testing of as many as 40% of asymptomatic individuals per work day (S4 Fig in S3 File).

A practical benefit of continuous testing strategies is that test results can be aggregated to derive an ongoing measure of prevalence in the workplace. If the employer closely follows such metrics, then mitigation strategies, such as augmentation of personal protective equipment and other NPI, can be introduced in a timely manner. To understand the impact of such interventions, we use our model to study how mitigations can impact the time dynamics of disease prevalence when they are introduced at a predetermined level of an "outbreak". In Fig 3, we show how the trajectory of disease spread can be altered for the "University" setting with

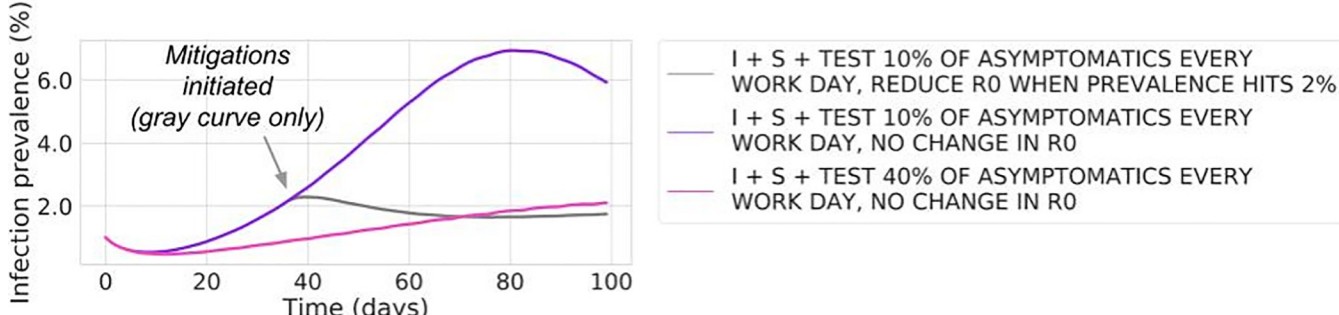

**Fig 3. Workplace prevalence trajectories in the "University" case study, where workplace transmission can decrease as a result of mitigations introduced upon an increase in prevalence.** Continuous testing allows an employer to estimate the current prevalence of infection in the workplace. This gives the employer the opportunity to introduce mitigations in response to increases in prevalence, reducing workplace spread. The gray curve shows the impact on workplace prevalence of testing 10% of asymptomatics per day but also reducing the workplace reproduction number (R0W, see S1 Table) from 4 to 2 (corresponding to initiating mitigations) when prevalence reaches 2%; note that prevalence trajectories are plotted as in Fig 2. The purple curve (10% asymptomatic testing with no change in R0W during the simulation) corresponds to the purple curve in Fig 2C, and the pink curve (40% asymptomatic testing with no change in R0W during the simulation) corresponds to the pink curve in Fig 2C.

a constant testing strategy (everyone tested approximately every 2 weeks), but where additional mitigations are initiated as a response to the prevalence reaching an "unacceptably high" level. We find that introducing mitigations at 2% prevalence can reduce peak prevalence from 6.8% (Fig 3, purple curve) to 2.3% (Fig 3, gray curve). Of note, this is roughly equivalent to the case where mitigations are not introduced but instead testing is performed at fourfold the level, i.e., everyone is tested approximately twice per week (Fig 3, pink curve; see also S4c Fig in S3 File).

This analysis emphasizes the two distinct benefits of continuous testing: (i) detection and isolation of infectious individuals, directly suppressing disease spread; (ii) use of aggregated test results to estimate infection prevalence in the workplace, allowing an outbreak to be recognized in its early stages so that mitigations can be rapidly deployed. As noted above, an "Office workplace" can maintain prevalence below 3% by testing its workforce approximately every 2 weeks, whereas a "University" without dynamic mitigations requires its population to be tested as much as fourfold as often to achieve that goal (pink curves in Figs 2C and 3). However, as shown in Fig 3, a "University" that performs ongoing monitoring testing and can quickly react to a growing outbreak at 2% can maintain a prevalence below 3% (Fig 3, gray curve), yet requiring only as much testing as an "Office workplace" that has the advantage of lower contact load and not having people living together full time.

The "University" setting modeled in the above simulations assumes a community population of 500,000. Thus, it is more relevant to a university campus in a medium to large city rather than to a small college town. To understand the impact of community size on transmission, we repeated the "University" simulation using a community population of only 3,000 people, while maintaining the campus ("workplace") population at 1,000. For this analysis, we assumed initial testing before return to campus, as well as ongoing testing of symptomatic individuals (but no randomized testing of asymptomatics). As shown in Fig 4A, the smaller community size does not have much impact on transmission on campus. However, it does cause a large increase in peak prevalence in the community (Fig 4B), from 2.8% with a community population of 500,000 to 12.8% for the community population of 3,000.

An advantage of analyzing results from the simulations performed herein is that we can easily tally the sources of infections, in contrast to real-world infections where it is quite difficult

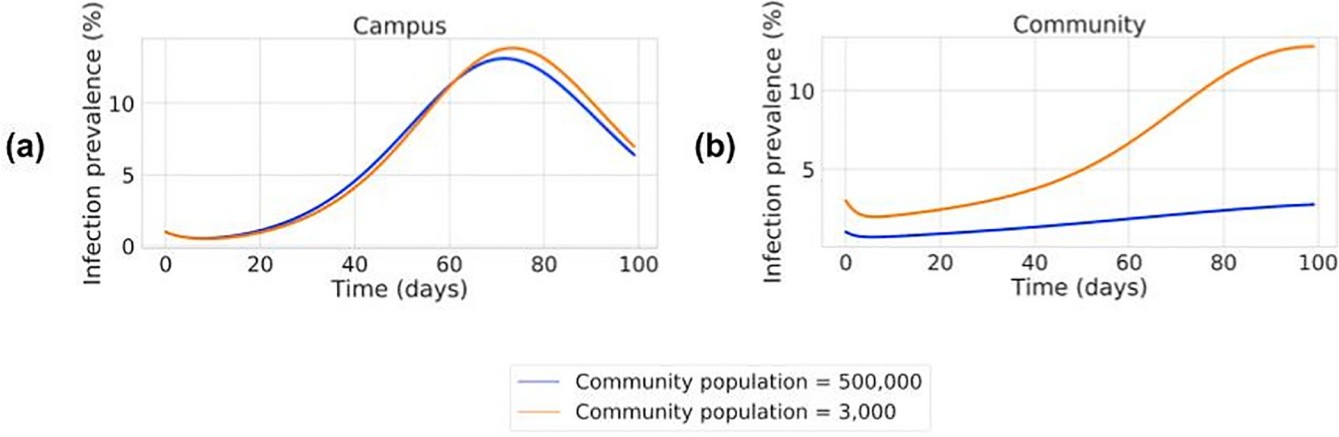

**Fig 4. Impact of varying community population size on the workplace and community prevalence trajectories in a "University" setting with no asymptomatic testing.** Prevalence trajectories (plotted as in Fig 2) for campus ("workplace", panel a) and community (panel b) populations, for scenarios where initial testing and symptomatic testing are performed in the "University" (I + SYMPTOMATIC TESTING). The size of the community population here is either 500,000 (as used in all other simulations, blue curves) or 3,000 (orange curves). The campus ("workplace") population is held constant at 1,000, as are all other simulation parameters as per S2 Table. Thus, the blue curve in panel (a) simulates the same scenario of campus prevalence for a "University" in a larger city, as in the green curve in Fig 2C.

to make such determinations. For the community population of 500,000, a total of 91,834 community members (18.4% of the community [S6c Fig in S3 File]) are infected during the simulation. We find that only 0.16% of these community infections are due to direct infection from the university population. In marked contrast, with the small community population of 3,000, a total of 719 community members (24.0%) are infected over the course of the simulation. Here, we find that 15.2% of these infections are directly attributable to disease transmission from the university. That is, these transmissions into the community arise from interactions between the infected university population and the susceptible community population, followed by community spread that ultimately results in a markedly increased community prevalence. Overall, these results illustrate the advantage of explicitly capturing the time dynamics of interactions between the community and the workplace.

## Discussion

Effective testing strategies are critical in allowing workplaces and schools of higher education to resume selected on-site activities, while minimizing the risk of outbreaks. The compartmental model presented here can be used to project how various testing strategies may impact the prevalence of infection in the workplace over time, allowing workplaces and schools to make more informed decisions about which testing strategy is best for them. Moreover, the model yields insights regarding when to introduce NPI mitigations when an outbreak is detected.

The compartmental model presented in this paper contains a SEPAYR model to track the epidemic in the community and includes interactions between this community population and the workplace population. In contrast, recent studies with similar goals have used single or continuous infection "seeds" to model infection originating from the community [3, 4, 8]. While such a simplification allows the impact of varying workplace parameters to be studied, it does not capture the impact of changes in community prevalence on the workplace. Modeling the community also allows the model to capture the impact of workplace spread on the surrounding community. This becomes particularly important in a setting such as a small "college town" (Fig 4), where the size of the "workplace" population (i.e., students and university staff) is of a similar magnitude to the size of the surrounding community [10]. More generally, because the safety of a community is dependent on lack of spread within its local workplaces, these results support the argument that it is in the interest of communities to support their places of business in preventing onsite spread of disease.

While the model presented in this paper gives a simplified view of the dynamics of COVID-19 infectiousness and transmission, the general framework provides a foundation for addressing real-world community-workplace scenarios. However, we emphasize that the assumptions inherent in this modeling approach must be clearly described to those who use its output to guide decisions. In S2 File, we review key considerations in understanding these parameters and some potential impacts of the values that are chosen.

Chief among such assumptions are the parameter choices that tailor the model to specific workplace settings. However, estimating the reproduction number ($R0_W$, see S1 Table) that will be experienced in a particular workplace setting is still an active area of research (see S2 File). Specifically, owing to the novelty of this virus, there is still only limited quantitative data on how and to what degree transmission occurs (e.g., by aerosol, contaminated surfaces, etc. [11]), especially in specific environments (e.g., in an air-conditioned office without fresh air but with physical distancing and mask wearing).

The uncertainty that exists regarding the degree of disease transmission needs to be considered in the selection of an appropriate testing strategy. In particular, in comparing the "Office workplace" and "Factory floor" simulations, we found that more aggressive testing strategies

control against outbreaks even in scenarios where there is higher contact in the workplace and thus diminish differences between higher and lower contact settings. Therefore, when keeping prevalence extremely low is critical to ensure employee health and continued business operations, choosing as aggressive a testing strategy as possible (under budgetary and logistic constraints) will give the business the best chance of continuing to operate even with worst-case transmission rates.

In addition to the uncertainty introduced by disease modeling, the optimal testing strategy for a specific workplace depends heavily on the cost-benefit analysis performed with that setting in mind. For example, reducing capacity in a factory setting may require shutting down a production line that would take several weeks to restart. This may lead to a higher peak prevalence being tolerated, as compared to an office workplace setting in which employees are able to smoothly transition between working at home and working in the office with very little disruption to company operations. As a result, this study does not aim to prescribe specific policy decisions that should be made in response to the model output, but instead presents a model that can be used by individual workplaces to understand the impact of different testing strategies through a lens specific to their use case.

The results presented here use test performance characteristics chosen to match PCR tests. Varying these parameters to instead mirror the performance of antigen tests or other rapid diagnostic tests could aid workplaces in determining the best test type to use in their testing program.

In this work, we are somewhat conservative in the outcomes of the modeling (i.e., may overestimate infections) by not explicitly accounting for testing of community members, or for any degree of self-isolation undertaken by community members who exhibit symptoms. Nevertheless, such factors are implicitly accounted for in the choice of community and workplace R0 values.

For simplicity, the compartmental model we present here assumes a homogenous workforce population. However, in many workplace settings, there are distinct subgroups of employees with varied behaviors that result in differential infection risk. For example, such subgroups may include employees who work in different locations, employees who are customer-facing vs. those who have very little interaction with others, or university faculty vs. students. Employers may choose different testing strategies for these subgroups. Thus, a natural extension of the modeling here would be to permit such models to capture heterogeneity among the employees. This would likely require the addition of a set of new compartments for each subgroup, as well as parameters describing the rates of infection between every pair of subgroups. These complexities may make agent-based modeling [12] better suited for this generalization. Similarly, generalizing to a heterogeneous community population would require the addition of analogous structure to the model. On a related point, age and other comorbidities have been shown to result in clinical heterogeneity once a person is infected; [13] this model could be extended to account for clinical outcomes such as hospitalizations or deaths.

One of the purposes of testing (both symptomatic and monitoring of asymptomatics) is to detect infected individuals and remove them from the workplace in order to prevent workplace-acquired infections. As described above, another critical benefit of testing is to leverage the aggregated test results to continuously estimate infection prevalence in the workforce. By assessing workplace safety in real time, actions can be taken to prevent emerging outbreaks from growing (Fig 3). Such actions may include changes to existing mitigation measures (such as enforcing that personal protective equipment is properly used), or even shutting down the workplace if prevalence exceeds unacceptable thresholds. To ensure that such actions are taken in a timely manner, but only if necessary, it is critical that workplace prevalence be watched closely. In practice, the results of monitoring testing need to be translated to an

estimate of workplace prevalence, with the statistical uncertainty around this estimate decreasing with larger sample sizes (i.e., with a larger volume of monitoring testing). Therefore, a return-to-work strategy that relies on quickly responding to nascent outbreaks benefits from a higher volume testing strategy that provides tighter statistical estimates of workplace prevalence. Such estimates would then be used to assess the safety of the workplace remaining open, for example, through either a formal statistical hypothesis test or by calculating a statistical distribution of likely prevalence values.

When deploying such testing programs in the real world, a key requirement for the employer is budgeting for the cost of the program despite the many uncertainties that the future holds. In our simulations, while the total number of tests performed varied substantially across testing strategies, the numbers for a particular strategy remained relatively stable across case studies (S7 Fig in S3 File). The main variation in the number of tests arises from testing of variable numbers of symptomatic individuals when outbreaks do occur. This relative stability of testing volume, for a fixed testing regimen, enables testing budgets to be estimated to a high degree of accuracy even before reopening a workplace, when parameters such as the reproduction number are still not known.

The results of this modeling show that, under the same testing conditions, higher peak prevalence is likely to be reached in settings with higher density and more interaction between workers, such as in a factory setting. Additionally, as described above, these workplaces are likely to have higher tolerance for peak prevalence before choosing to shut down operations, due to the higher costs associated with shutting down and restarting sections of the factory. These results have clear implications for increasing social inequalities in health, which should be carefully considered by decision-makers at the workplace level, but also by local and state governments. Ensuring that testing for COVID-19 is readily accessible outside of a workplace setting may help to lessen the disparate impact of the virus across different social groups.

Repeated assessment of model adequacy is expected to be necessary due to the relatively short time horizon for which predictions can be reliably trusted due to the ever-changing state of societal policies and behaviors, as well as evolving clinical knowledge of the disease. Model parameters should be continuously updated to reflect scientific understanding of the disease, and also to reflect the observed test results and symptom reporting from the workplace of interest. This will allow the model output to be used to provide ongoing guidance about the projected impact of different workplace testing and other mitigation strategies. A concrete example of implementing such updates is the real-time estimation of the community virus reproduction number, $R0_C$ (see S2 File). The estimation of this reproduction number is based on data across all COVID-19 variants, so should capture the impact of new variants that are more transmissible. Similarly, since the impact of the vaccination rollout will be seen in the number of cases reported, the model can capture the impact of vaccinations without them being explicitly modeled.

## Conclusion

We present an epidemiological compartmental model that demonstrates the impact of testing strategies and dynamically-introduced employer mitigations on the spread of COVID-19 in a workplace. This model captures interactions between the workplace and the surrounding community population and can be tailored to fit the specifics of a wide range of workplace scenarios. To illustrate this flexibility, we present three case studies, simulating an office workplace, a factory floor, and a university campus. We discuss how to interpret insights from these simulations and how this model can guide the volume of testing intended to prevent workplace outbreaks from occurring or becoming large.

We also show how this modeling approach can allow employers to quantify how using ongoing testing can inform the real-time introduction of mitigations intended to prevent disease spread when outbreaks begin. In particular, we find that pairing data-driven mitigations with ongoing testing in a university can achieve the same benefit as substantially more testing. Additionally, we demonstrate how modeling the workplace and community populations together allows us to uncover important dependencies between these populations, which are particularly acute when the size of these populations is similar. In this setting, an outbreak in the workplace can lead to increased infection in the community, even when the community itself has mitigations in place to reduce transmission. Lastly, we reiterate that data from the workplace of interest should be used to adjust model parameters over time. This approach should improve model accuracy for continuous forecasting of disease prevalence and thus better empower employers to choose testing strategies that meet the goal of keeping their business up and running within explicit safety parameters.

## Supporting information

**S1 File. Supplementary methods.**
(ZIP)

**S2 File. Estimation of community and workplace model parameters.**
(DOCX)

**S3 File. Case study results.**
(ZIP)

**S4 File. References.**
(DOCX)

**S1 Table. Notation and formulae for parameters used in the "community-workplace" model.**
(PDF)

**S2 Table. Model parameters held constant for all case studies.** All model parameters not listed here or in the main text can be calculated from these values using the formulae in S1 Table. See S2 File for a discussion of how these values were selected.
(PDF)

## Author Contributions

**Conceptualization:** Sarah F. Poole, Jessica Gronsbell, Alexander R. Macalalad, Nigel Snoad, Michael P. Brenner, Robert M. Califf, Menachem Fromer.

**Formal analysis:** Sarah F. Poole, Jessica Gronsbell, Dale Winter, Stefanie Nickels, Roie Levy.

**Investigation:** Sarah F. Poole, Jessica Gronsbell.

**Methodology:** Sarah F. Poole, Jessica Gronsbell, Dale Winter, Stefanie Nickels, Roie Levy, Bin Fu, Maximilien Burq, Sohrab Saeb, Matthew D. Edwards, Vignesh Kumaresan, Alexander R. Macalalad, Nigel Snoad, Michael P. Brenner, Paul Varghese, Robert M. Califf, Vindell Washington, Vivian S. Lee, Menachem Fromer.

**Project administration:** Sneh Shah, Michelle Prevost, Nigel Snoad.

**Software:** Sarah F. Poole, Jessica Gronsbell, Stefanie Nickels, Roie Levy, Bin Fu, Sohrab Saeb, Michael K. Behr.

**Supervision:** Michael P. Brenner, Lance J. Myers, Paul Varghese, Robert M. Califf, Vindell Washington, Vivian S. Lee, Menachem Fromer.

**Validation:** Sarah F. Poole, Jessica Gronsbell, Maximilien Burq, Sohrab Saeb, Matthew D. Edwards, Alexander R. Macalalad.

**Visualization:** Sarah F. Poole, Jessica Gronsbell.

**Writing – original draft:** Sarah F. Poole, Jessica Gronsbell, Menachem Fromer.

**Writing – review & editing:** Sarah F. Poole, Jessica Gronsbell, Dale Winter, Stefanie Nickels, Roie Levy, Bin Fu, Maximilien Burq, Sohrab Saeb, Matthew D. Edwards, Michael K. Behr, Vignesh Kumaresan, Alexander R. Macalalad, Sneh Shah, Michelle Prevost, Nigel Snoad, Michael P. Brenner, Lance J. Myers, Paul Varghese, Robert M. Califf, Vindell Washington, Vivian S. Lee, Menachem Fromer.

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
