## [Decision Letter · Decision Letter 0]

27 Apr 2021

PONE-D-20-40468

A holistic approach for suppression of COVID-19 spread in workplaces and universities

PLOS ONE

Dear Dr. Poole,

Thank you for submitting your manuscript to PLOS ONE. After careful consideration, we feel that it has merit but does not fully meet PLOS ONE’s publication criteria as it currently stands. Therefore, we invite you to submit a revised version of the manuscript that addresses the points raised during the review process.

Your manuscript was reviewed by one expert in the filed. Many potential reviewers were invited but could not accept invitation to review. However, it seems that the reviewer captured most important points that need your attention. Please carefully consider the attached comments and provide point-by-point responses.

We look forward to receiving your revised manuscript.

Kind regards,

Yury E Khudyakov, PhD

Academic Editor

PLOS ONE

Journal Requirements:

1. Please ensure that your manuscript meets PLOS ONE's style requirements, including those for file naming. The PLOS ONE style templates can be found athttps://journals.plos.org/plosone/s/file?id=wjVg/PLOSOne_formatting_sample_main_body.pdf and https://journals.plos.org/plosone/s/file?id=ba62/PLOSOne_formatting_sample_title_authors_affiliations.pdf

2. Please provide more information on the nature of your competing interests, e.g. clarify whether commercial interests exist related to the current study.

'At the time of work, SFP, JG, DW, SN, RL, BF, MB, SS, MDE, MKB, VK, ARM, SS,

MP, NS, LJM, PV, RMC, VM, VSL, and MF were employees of and owned equity in

Verily Life Sciences. While this manuscript does not explicitly mention Verily's Healthy

At Work program, the models presented herein are used in that program.'

We note that one or more of the authors are employed by a commercial company: Verily Life Sciences & Google Research.

Additional Editor Comments (if provided):

Reviewers' comments:

Reviewer's Responses to Questions

**Comments to the Author**

1. Is the manuscript technically sound, and do the data support the conclusions?

Reviewer #1: Yes

2. Has the statistical analysis been performed appropriately and rigorously? 

Reviewer #1: Yes

3. Have the authors made all data underlying the findings in their manuscript fully available?

Reviewer #1: Yes

4. Is the manuscript presented in an intelligible fashion and written in standard English?

Reviewer #1: Yes

5. Review Comments to the Author

Reviewer #1: This seems like an interesting model that is well described by the authors (I particularly appreciate all the supplementary files and tables describing the parameters). I do have a few comments related to the contents.

Major comments

There are many policy implications for your model's test results, yet you do not present them clearly in the Discussion. What are the implications of your findings in terms of stepping up testing capacities and diversifying testing tools (e.g., using rapid diagnostic testing for instance)? In addition to such practical policy implications, the authors do not make a case for considering social inequalities in health, even though their differentiated model (according to the type of workplace: factory floor vs. office space; and university) implicitly highlights such inequalities. Given that COVID-19 exacerbates such social inequalities in health (e.g., people working in factories are more likely to be infected with COVID-19), based on your model testing, what conclusions can you draw?

I am sure the authors could also account for current and (modelled) upcoming major changes in transmission data, particularly in the context of the spread of new variants of SARS-CoV-2 (based on the available clinical knowledge that points to increased infection rates). This new data will surely impact your modelling and should therefore be taken into consideration through additional scenarii.

Minor comments

Blue curve details (which I believe stands for 'no testing') is missing from Figure 2.

Please refer to physical distancing rather than 'social distancing'.

6. PLOS authors have the option to publish the peer review history of their article (what does this mean?). If published, this will include your full peer review and any attached files.

Reviewer #1: No

---

## [Author Response · Author response to Decision Letter 0]

28 May 2021

Thank-you for the review of our manuscript entitled ‘A holistic approach for suppression of COVID-19 spread in workplaces and universities’, and the opportunity to resubmit our revised manuscript. 

We appreciate the careful review of our manuscript, and the thoughtful comments around implications of the model results. We have added to the manuscript to acknowledge these implications.

The reviewer mentioned the many policy implications for our model’s test results, including on increasing testing capabilities. The goal of our manuscript is not to prescribe an optimal testing strategy, as through our work with various employers we have found that the cost-benefit trade off being performed varies significantly across different industries and, more generally, across different employers. We have added commentary in the Discussion section to note that our focus is on presenting a model for others to use and interpret according to their specific use case, rather than on presenting specific model results that prescribe the best decisions to make moving forward. We hope that this addresses the reviewer’s point satisfactorily. 

The reviewer also mentioned that the model results may have implications for diversification of testing tools, such as the use of rapid diagnostic testing. The model that we present has a large number of parameters to explore, and to keep the scope of the manuscript manageable we have chosen to focus on specifically varying the parameters that define the characteristics of the workplace environment, and on the frequency of testing. We agree with the reviewer’s insight that this model could be used to explore the impact of different testing types, and have added this as a note in the Discussion section to highlight this use of the model. 

The reviewer notes that the model results highlight social inequalities in health, in that it shows a higher peak prevalence is likely in an employee population that works in a factory setting compared to an office workplace. We agree that the disparate impact of the virus across social groups is of key importance and concern. We have added language in the Discussion section to acknowledge this result, and highlight that this should be carefully considered by groups determining the appropriate testing strategy to employ. We have also noted the importance of testing availability outside a workplace setting (i.e., testing provided by the state or county) in ensuring that all members of the community are able to monitor their health. We thank the reviewer for highlighting this important point, and hope that the insights provided in our manuscript can help both workplaces and policy makers to make informed decisions about how their testing programs will impact the population that they are serving

Another point that was raised by the reviewer is that the emergence of new viral variants, with different transmission characteristics, will impact the model results. We agree that results will certainly be impacted by changes in transmissibility of the virus, and also note that the vaccination rollout will impact the spread of the virus. The main parameters in the model that captures the virus transmissibility are R0W and R0C. As we discuss in Supplement S2, the value used for RoC is obtained from publicly available modeling that identifies the value that best fits the trend in the number of reported cases. Both new variants and vaccination amount will impact the number of cases being reported, so we believe that the model is able to capture the impact of these key changes without them being explicitly included in the transmission model. 

We also thank the reviewer for identifying the missing legend label for Figure 2. This has been updated. Additionally, we have updated our language to more accurately refer to ‘physical distancing’ rather than ‘social distancing’. 

Finally, we have updated our manuscript formatting and file naming to better comply with requirements. We have also amended our Funding Statement and Competing Interests Statement, and these are included in the Cover Letter, as requested.

---

## [Editor Report · Decision Letter 1]

5 Jul 2021

A holistic approach for suppression of COVID-19 spread in workplaces and universities

PONE-D-20-40468R1

Dear Dr. Poole,

We’re pleased to inform you that your manuscript has been judged scientifically suitable for publication and will be formally accepted for publication once it meets all outstanding technical requirements.

Kind regards,

Yury E Khudyakov, PhD

Academic Editor

PLOS ONE
---

## [Editor Report · Acceptance letter]

27 Jul 2021

PONE-D-20-40468R1 

A holistic approach for suppression of COVID-19 spread in workplaces and universities 

Dear Dr. Poole:

I'm pleased to inform you that your manuscript has been deemed suitable for publication in PLOS ONE. Congratulations! Your manuscript is now with our production department. 

Kind regards, 

on behalf of

Dr. Yury E Khudyakov 

Academic Editor

PLOS ONE